# The Incidence and Severity of Post-Vaccination Reactions after Vaccination against COVID-19

**DOI:** 10.3390/vaccines9050502

**Published:** 2021-05-13

**Authors:** Izabela Jęśkowiak, Benita Wiatrak, Patrycja Grosman-Dziewiszek, Adam Szeląg

**Affiliations:** Department of Pharmacology, Faculty of Medicine, Wroclaw Medical University, Mikulicza-Radeckiego 2, 50-345 Wrocław, Poland; benita.wiatrak@umed.wroc.pl (B.W.); patrycja.grosman-dziewiszek@umed.wroc.pl (P.G.-D.); adam.szelag@umed.wroc.pl (A.S.)

**Keywords:** vaccination, COVID19, adverse reaction after vaccination

## Abstract

The pandemic of COVID-19 might be limited by vaccination. Society should be vaccinated to prevent the spread of coronavirus disease 2019 (COVID-19) and to protect persons who are at high risk for complications. In Poland, the National Vaccination Program has been introduced, which is a strategy for planning activities to ensure safe and effective vaccinations among Polish citizens. It includes not only the purchase of an appropriate number of vaccines, their distribution but also monitoring of the course and effectiveness of vaccination and the safety of Poles. The national COVID-19 immunization program has been divided into four stages. Stage 0 covers the healthcare workers to be vaccinated first, as they are most at risk of being infected with the coronavirus. The study aims to prove the thesis that GIS statistical data on the incidence of COVID-19 post-vaccination reactions should be verified, as patients do not report their occurrence through the procedure indicated by GIS. In March 2021, an anonymous questionnaire survey was conducted using an electronic questionnaire among persons belonging to group zero of the National Vaccination Program. The survey consisted of 19 short questions concerning, inter alia, getting COVID-19, post-vaccination reactions after receiving the first and second doses of the COVID-19 vaccine, and motivation to proceed with vaccination. A total of 1678 complete responses were received. It has been shown that only a small number of post-vaccination reactions are reported to the Sanitary Inspection, which makes GIS statistics on the incidence of post-vaccination reactions in COVID-19 unreliable. In addition, having earlier suffered from COVID-19 had an impact on the occurrence of more severe side effects after the first dose of the COVID-19 vaccine.

## 1. Introduction

The novel coronavirus 2019 (SARS-CoV-2) has caused the worldwide pandemic of coronavirus disease 2019 (COVID-19) [1]. The disease is asymptomatic or mild in most patients. However, a substantial percentage of people have more extensive pneumonia that can progress to hypoxaemic respiratory failure, shock, dysfunction of organs and death [2]. The risk factors for severe COVID-19 include being male, older age, and having lung disease, cardiovascular disease, obesity, diabetes and hypertension [3,4,5,6]. Unfortunately, no effective treatment has been demonstrated to radically change the natural history of SARS-CoV-2 infection [7].

As immunization is one of the most successful and cost-effective health interventions to prevent infectious diseases, vaccines against COVID-19 are considered to be of great importance to prevent and control COVID-19 [8]. It is, therefore, important to vaccinate against COVID-19 in most of the population, as this can have large consequences for the success of a vaccination program—with potentially large health and economic consequences. [9] Clinical trials support the safety and efficacy of vaccines developed by Oxford-AstraZeneca and Pfizer [10,11].

In Poland, according to data as of 18 February 2021, after administration of 2,384,794 doses of vaccines, 2131 adverse vaccine reactions were reported to the Chief Sanitary Inspector (source: www.gov.pl/web/szczepimysie (accessed on 18 February 2021)), which suggests that they occur very, very much rarely, only 1 in 1100 administrations of the vaccine. Meanwhile, our observations contradict these data and suggest that vaccination symptoms are very common, although their severity varies. In addition, a review of the scientific literature shows that the currently published COVID-19 vaccine surveys are mainly concerned with assessing the number of respondents ready for vaccination, but there are no studies on COVID-19 disease, post-vaccination reactions after intake of the first and second doses of COVID-19 vaccine, as well as the motivation for vaccination.

The preparation is the first vaccine against COVID-19 approved for use in the European Union from Pfizer/BioNTech. As expected, the largest number of people from the vaccination program group zero was given this vaccine, which was also the first one available in Poland. Preparation of Pfizer/BioNTech’s two-dose intramuscular vaccine indicated that in winter it must be given 21 days after receiving it. The vaccinated mRNA contains the messenger ribonucleic acid (mRNA) encoding the S (spike) protein of the SARS-CoV-2 virus. In the host cell, the SARS-CoV-2 S (spike) protein is synthesized, which, being a strong antigen, stimulates the immune response in the form of neutralizing antibodies (humoral response) and stimulates the production of T lymphocytes (cellular response).

This study aimed to verify the actual incidence of vaccine reactions during vaccination against COVID-19. The surprising fact is that there were few reports of post-vaccination reactions through the procedure indicated by GIS. In addition, another goal of the study was to assess whether a previous COVID-19 incidence has an impact on the occurrence of certain symptoms and their severity, as well as their frequency, after receiving the first and/or second dose of the COVID-19 vaccine.

## 2. Materials and Methods

### 2.1. Study Design, Population and Sampling

An anonymous study in the form of an electronic questionnaire was conducted among people belonging to group zero of the National Vaccination Program. In March 2021, electronic questionnaires were sent to employees and students of the Medical University in Wrocław, as well as to District Pharmaceutical Chambers and District Medical Chambers. The Supreme Medical Chamber gives the total number of doctors, amounting to 181,002 people as of 8 April 2021, while the Supreme Pharmaceutical Chamber gives the total number of pharmacists 33,988. It is difficult to determine to what extent all district medical and pharmacy chambers distributed the questionnaire. Moreover, at the Wroclaw Medical University, there are 2345 academic teachers employed, to whom the questionnaires were sent. Similarly, questionnaires were sent to students (5616) of the Wroclaw Medical University.

The questionnaire google provides online questionnaire design and survey functions for enterprises, research institutions and individuals. The survey consisted of 19 questions. A total of 1249 people completed the questionnaire. The study was approved by the Bioethics Committee of the Wrocław Medical University (KB209/2021). In Poland, after 14 January 2021, there was a change in the composition of the zero group. Before that date, administrative employees of medical universities could get vaccinated.

### 2.2. Measures

The questionnaire included questions about belonging to the group determining vaccination at stage 0 (academic teacher employed at a medical university, doctor, pharmacist, pharmaceutical technician, nurses and midwives, a doctoral student at a medical university, a medical university student and others. The group “others” includes laboratory diagnosticians, physiotherapists, technicians and administrative staff).

The content of the questionnaire also included (1) social characteristics, such as education (higher, secondary, PhD student, student, vocational), gender (female, male), age (divided into groups 19–30, 31–40, 41–50, 51–60 and over 60 years of age)—a graphic summary presented in Figure 1; (2) regarding COVID-19, such as being test-confirmed and unconfirmed with COVID-19, determined on the basis of symptoms and circumstances, and the course of COVID-19 (asymptomatic, mild—mild upper respiratory tract complaints, elevated body temperature not exceeding 38 °C, cough or shortness of breath, treatment at home, without contacting a doctor, moderate-fever > 38 °C, persistent dry cough, dyspnoea, pulmonary involvement visible in imaging tests, outpatient treatment in contact with a doctor, severe-hospital treatment, very severe-hospital treatment with the use of a ventilator); (3) factors influencing respondents’ decisions about vaccination against COVID-19; (4) post-treatment reactions to vaccination symptoms such as symptoms occurring after the first and second dose of the vaccine, comparison of the severity of symptoms after the first and second dose of the vaccine, and reporting the post-vaccination reaction to the Sanitary Inspection; (5) acceptance, attitude and vaccination preferences for future COVID-19 vaccinations; and (6) COVID-19 vaccine preparations.

The self-reported questionnaire was designed, among others, based on the Adverse Vaccine Reaction Reporting Card, the Summary of Product Characteristics of Pfizer vaccine, which was mainly vaccinated in group zero, and other adverse reactions.

Different numbers of people answering given questions result from the possibility of opting out of the questionnaire from answering the question.

### 2.3. Statistical Analysis

Statistical analyses were performed with Statistica v13.0. Pearson’s chi-square test was used to compare the differences between the different subgroups.

## 3. Results

### 3.1. Study Sample Characteristics

The largest number of people who took part in the study belonging to the group zero of the National Vaccination Program against COVID-19 were students of medical schools (611), followed by pharmacists (521), doctors (266) and academics employed at a medical university (198). In addition, the study group consisted of a smaller number of nurses (41), pharmaceutical technicians (26), doctoral students at a medical university (22) and other people (including laboratory diagnosticians, midwives, physiotherapists, technicians and administrative staff).

According to the Central Statistical Office (GUS), on 31 December 2019, 65.3 thousand people worked in public pharmacies, company pharmacies and pharmacy outlets, including 26.1 thousand pharmacy masters and 33.3 thousand pharmaceutical technicians. The vast majority of the employed were women (almost 83% of pharmacy masters and over 95% of pharmaceutical technicians). A total of 521 pharmacists took part in the study, which is approx. 2% of the entire population of pharmacists. In the study, there is a visible advantage of women (1301) over men (355), which is generally due to the quantitative advantage of women employed in the health care sector compared to men.

Since the beginning of the 1990s, the level of education of society has been increasing at an extremely fast pace, especially in the case of women. According to statistics kept by the Central Statistical Office of Poland (GUS), among the total number of people with higher education, the share of women increased in the period 1988–2002 from 47.0% to 54.8%. Most people who took part in the study had higher education, as much as 63%—1044 persons (secondary education—212,382 students and 17 doctoral students). There were discrepancies in the selection of the level of education. This may be because the question about belonging to group zero of the National Vaccination Program was a multiple-choice question, and the question about education was characterized by a single choice. The age distribution of the respondents was even. Most people (852) who took part in the survey are aged 19–30, which constitutes 51.4%. From the remaining age groups, 20.5% (340) are people in the 31–40 age group, 12.1% (201) in the 41–50 age group, 8.9% (148) in the 51–60 age group and 6.9% (115) over 60 years of age. 

Moreover, many people from the studied group zero of the National Vaccination Program have more than one profession or are still undergoing other studies, as indicated by the results obtained when analyzing the question about belonging to group zero.

### 3.2. Incident of COVID-19

A total of 195 people, i.e., 11.8%, suffered from COVID-19 confirmed by the test, while 83.3% (1385) did not suffer from it, and 72 people refrained from selecting the answer. However, based on the circumstances and symptoms, 18.5% (299) of people developed COVID-19, unconfirmed by the test, and 72.5% (1173) of people did not suspect COVID-19, but the asymptomatic transition of COVID-19 cannot be ruled out. When asked about the course of COVID-19, the answers were obtained that 74 people had an asymptomatic course, 325 people mild, 69 people moderate, and 1185 people did not suffer from COVID-19. A total of 468 suffered from COVID-19, which does not coincide with the responses to the previous question, i.e., people with a confirmed COVID-19 test result and confirmed based on symptoms and circumstances, i.e., 488 people. Assuming that 468 people suffered from COVID-19, it can be stated that as many as 28.3% of the respondents suffered from COVID-19.

### 3.3. Impact Factors of the COVID-19 Vaccined

In a multiple-choice closed question, the respondents chose what factors enabled them to decide to vaccinate. Most people, as many as 80% (1327), decided to vaccinate against COVID-19 because of concern for the health of their relatives. Fewer respondents, 76% (1256), decided about vaccination due to concerns about their health and 66% (1099) because of knowledge about COVID-19 related to their occupation, 56% (927) due to the presence of a pandemic and 49.1% due to the possibility of traveling. In this question, 50 people indicated that they did not agree to be vaccinated against COVID-19, and 5 people did not consent to any voluntary vaccination.

The respondents considered the encouragement of scientific and medical authorities—40% (653), the presence of chronic diseases—9% (153), encouragement from the employer—3.4% (57), media information, including advertisements—2.8% (47) and severe COVID-19 history in the past—2.5% (39).

### 3.4. Adverse Reaction after Vaccination

For the largest number, 78% (1253), of respondents who received the first dose of the vaccine, the most common side effect was soreness at the injection site. Limb pain was reported to a lesser extent with 46.6% (746). In contrast, less than 50% of respondents after the first dose of the COVID-19 vaccine experienced fatigue 30% (490), injection site swelling 24.5% (392), malaise 21.3% (342), injection site redness (295), headache (268), muscle and joint pain (240), elevated body temperature up to 38 °C (220) and chills (186). Other possible side effects after the first dose include injection site pruritus (89), lymph node pain and enlargement (68), fever above 38 °C (60), seizures (26), insomnia (69), nausea (64), vomiting (13), allergic reactions (11), migraine (43), diarrhea (22), cough (16), fainting (10) and hair loss (13).

After the second dose of the COVID-19 vaccine, the majority of respondents also experienced soreness at the injection site—64.7% (1008), in a smaller group, fatigue—45.7% (712), malaise (671), pain in the limb (604), muscle and joints (515), chills (483), headache (481), body temperature elevated up to 38 °C (446), injection site swelling (317), injection site redness (259), fever above 38 °C (223), lymph node soreness and enlargement (147), insomnia (109), nausea (106), injection site pruritus (76), migraine (54), seizures (50), cough (30), diarrhea (27), fainting (20), vomiting (18), hair loss (15) and allergic reaction (9), but 187 people had no adverse effects. In the next question, the respondents rated the severity of the side effects after receiving the first and second doses of the COVID-19 vaccine. They found that almost 60% of people (906) had more vaccine symptoms after taking the second dose, and 23% (350) had weaker symptoms after taking the second dose. However, no differences were found in 17.4% (265) of respondents.

Side effects after the second dose of the COVID-19 vaccine were mainly fatigue—31.8% (452), malaise (442), muscle and joint pain (345), chills (330), and body temperature increased to 38 °C (323), headache (307), injection site soreness (268), pain in a limb (240), fever above 38 °C (186), lymph node soreness and enlargement (87), injection site swelling (73), nausea (71), insomnia (61), injection site redness (51), seizures (45), migraine (43), injection site pruritus (24), cough (18), diarrhea (17), vomiting (15), fainting (13) and allergic reactions (5).

The range of side effects after vaccination is wide and affects 80% of vaccinated people. According to the results of the questionnaire, only 4.6% (73) of vaccinated people reported an adverse reaction after receiving the COVID-19 vaccine to the Sanitary Inspection. However, 1503 people, i.e., as much as 95.4%, did not report any side effects after taking the vaccine. This question was answered in the survey by 1576 people, while the number of completed questionnaires was 1657, which indicates that 81 respondents did not answer this question. 

We also consider responses to other questions in the COVID-19 vaccination survey. According to the responses to the survey, 50 people did not agree to be vaccinated against COVID-19, and 5 people did not agree to any vaccination. Side effects after the first dose of the vaccine did not occur in 121 people and in 187 people after the second dose of the vaccine. With this assumption, the side effect should not be reported by 176 respondents, and the rest, i.e., 1481 people, should report the side effect after vaccination. Of this group, only 73 people reported an adverse reaction, which means that the remaining 95% did not report an adverse reaction.

### 3.5. Acceptance, Attitude and Vaccination Preferences for Future COVID-19 Vaccinations

When asked how the respondent assesses the need for vaccination against COVID-19, 76.2% (1263) answered definitely yes, 18.8% (312) rather yes, 3.5% (58) rather not and 1.4% (24) not. On the other hand, when asked about satisfaction with the decision to vaccinate, 98% (1583) expressed satisfaction, and 2% (33) were dissatisfied with the decision, while 41 refrained from answering. There were discrepancies in the answers to this question, as it is known from previous answers to the questions that 55 people did not agree to be vaccinated against COVID-19.

The survey also asked about getting vaccinated again against COVID-19. This question was answered by 1625 respondents, 74% of whom (1202—yes) would like to be re-vaccinated against COVID-19, 23.1% (375—don’t know) are not sure and 3% (48—no) will not be vaccinated, and 32 people abstained from answering.

### 3.6. COVID-19 Vaccine Preparations

The question about what preparation the vaccinated respondents were with was answered by 1577 people out of 1657 respondents (80 people did not answer this question). Most of the subjects from group zero, as expected, were vaccinated with the preparation of the company Pfizer 94.9% (1497), AstraZeneca 3.9% (61) and Moderna 1.2% (19). When asked with the option to add another proposed answer about the possibility of choosing a vaccine preparation, 90.2% (1429) would choose the Pfizer vaccine, 3.8% (60) Moderna, 1.3% (20) AstraZeneca. This question was answered by 1585 respondents out of 1657 people who filled out the questionnaire. There were also responses that “I would get vaccinated with any single-dose vaccine” (12), or “it didn’t matter” (35). When asked what technology the vaccine against COVID-19 is based on, 1595 respondents were answered that the Pfizer and Moderna vaccine is an mRNA vaccine on the S protein, and vaccination with this preparation in this technology marked 89.3% (1425). The previous question shows that 1513 people were vaccinated with these preparations. The AstraZeneca vaccine is a vector vaccine based on the adenovirus S protein, and vaccination with the preparation obtained in this technology was indicated by 68 people (4.3%) in this question, while in the closed question about vaccination with AstraZeneca preparation, 61 people indicated vaccination with this preparation. In addition, 6.4% of people, or 102 people, do not know what technology the vaccine was based on. A total of 1596 people answered this question, while 1602 people were vaccinated, so 6 people abstained from answering.

A description of statistical significance and interpretation can be found in the Discussion Table 1. 

## 4. Discussion

The obtained research results indicate, first of all, the lack of reporting of side effects after the administration of the COVID-19 vaccine by people from group zero of the National Vaccination Program. After receiving the vaccine, the adverse reaction should be reported to the Sanitary Inspection via the application: salon.gov.pl or, only if it is impossible to report adverse reaction after vaccination via the above-mentioned application, on the Adverse Vaccine Reaction Reporting Card (NOP), i.e., according to the current methods of reporting NOP, but no later than 31.12.2021. The obligation to notify the NOP results from Art. 21 of the Act on preventing and combating infections and infectious diseases in humans, and failure to comply with it may result in Art. 52 fine for failing to report an adverse vaccine reaction. The State Sanitary Inspectorate collects epidemiological data on the occurrence of NOP based on information from reports and conducted epidemiological investigations and submits a copy of the NOP notification to the Office for Registration of Medicinal Products, Medical Devices and Biocidal Products (URPLWMiPB), which may order additional tests to be carried out on the vaccine or withhold/withdraw its approval for trading, and to the National Institute of Public Health—National Institute of Hygiene (NIZP-PZH), which is a national unit of expertise in the field of epidemiological supervision. From the information available at https://www.gov.pl/web/szczepimysie/raport-szczepien-przałko-covid-19, as of 24 March 2021, the number of vaccinations with the second dose is 1,808,008, and the number of post-vaccination reactions is 5189. Certainly, how the conducted study shows that the low number of side effects after vaccination is due to the failure to report side effects to the Sanitary Inspection.

By the accepted definition of NOP, we refer to a health disorder that occurs within 4 weeks after the administration of the vaccine. The statistically significant side effects after receiving the vaccine include pain at the injection site (1275; *p* < 0.0001), redness at the injection site (696–after the first dose; *p* < 0.0001) and pain in the limb (766 *p* < 0.0001) after the first dose of the vaccine, and after the second dose of the vaccine—a temperature above 38 °C (226 *p* = 0.04). The COVID-19 vaccines can cause mild adverse effects after the first or second dose, including pain, redness or swelling at the site of vaccine shot, fever, fatigue, headache, muscle pain, nausea, vomiting, itching, chills and joint pain, and can also rarely cause anaphylactic shock [12]. One woman interviewed reported having experienced a three-week pregnancy loss after receiving the first dose of the Pfizer vaccine. Phase 3 clinical trials assessing the efficacy and safety of the preparation are continued. 

There is a significant statistical association between the development of COVID-19 and the occurrence of stronger side effects after the first dose of the vaccine (103), and in those who had not previously had COVID-19, stronger side effects occurred after the second dose of the vaccine (40) (*p* < 0.0001). T-cell and antibody responses correlate with the severity of COVID-19 clinical disease. Those who previously had COVID-19 had stronger adverse effects after the first dose of the vaccine against COVID-19, which could be partly dependent on weakened antibody-dependent enhancement (ADE) (related to the more colorful name of cytokine storm). ADE corresponds to a situation where antibodies that normally alleviate the consequences of a viral infection end up doing the opposite: they fail to control the virus’ pathogenicity by failing to be neutralizing (i.e., the antibodies are not able to kill the virus), or even enhance its virulence either by facilitating its entry into the cell (thus enhancing the viral reproduction potential) or by triggering an extensive and misadapted reaction, thereby causing damage to the host organs through hyper-inflammation (cytokine storm) [13]. Patients with the milder disease also show greater clonal expansion and less active proliferation in CD8 T-cells in bronchial fluid as well as lower serum cytokine levels, compared to patients with severe disease [14] (Hellerstein 2020).

In the course of the research, a statistically significant correlation (*p* = 0.04) was found that women (238) suspected COVID-19 much more often than men (65). Moreover, women (1243) feel the need to vaccinate more often than men (318) (*p* = 0.043) and are more satisfied with the decision to vaccinate (women = 1251, men = 337; *p* = 0.043). People with higher education knew the technology of the vaccine with which they were vaccinated (*p* = 0.002) and they would choose a vaccine based on the VIRUS mRNA technology (Pfizer—(911) and Moderna (39)). In addition, people with higher education in the National Vaccination Program group zero also showed a statistically significant need to be vaccinated against COVID-19 (*p* = 0.036). 

The public debate on SARS-CoV-2 vaccination has been performed and used an online survey study among Polish healthcare workers from 22 December 2020 to 8 January 2021. The respondents were divided into two main groups: HCW and control (CG). HCW group comprised physicians (MD) and administrative healthcare assistants (HA). More than 94% of physicians were willing to be vaccinated against SARS-CoV-2. The main concern regarding vaccination in all groups proved to be the long-term side effects of the vaccine [15]. In the previous study by this research group, which was performed on medical and non-medical students, the analogous questionnaire was available from 22 December 2020 to 25 December 2020. An online survey was performed among Polish medical 687 (MS) and non-medical students 1284 (NMS; control group). Doctors showed more willingness to be vaccinated against COVID-19 (94.44%) and less fear of vaccine side effects than medical students. What is more, it was observed that with the increasing year of medical studies, the willingness to get vaccinated also increased [16].

A cross-sectional self-administered anonymous questionnaire survey was conducted among nurses in Hong Kong, China, on 26 February and 31 March 2020 to evaluate their acceptance of COVID-19 vaccination. In this study, responses from 806 participants were retrieved, where 40.0% of participants intended to accept COVID-19 vaccination. The publication was received by the editorial office of the journal in May 2020 [17].

Another survey of nurses in Hong Kong was conducted. A total of 1205 eligible nurses were included in the analysis. Nurses did not express their reluctance to vaccinate against COVID-19. The publication was received by the editorial office of the journal in September 2020 [18]. In our studies, all included nurses expressed a strong desire to be vaccinated against COVID-19 (33). On the other hand, a nationwide cross-sectional, self-administered online survey was conducted on 1–19 May 2020 in China. A total of 3541 complete responses were received. The majority reported a probably yes intent (54.6%), followed by a definite yes intent (28.7%) to be vaccinated against COVID-19 [19]. What is more, over half of U.S. firefighters and EMS workers were uncertain or reported low acceptability of the COVID-19 vaccine when it becomes available. A cross-sectional study by using an anonymous online survey was conducted among the 3169 respondents in October 2020 [20]. 

The nationwide online survey of 804 U.S. English-speaking adults indicated that COVID-19 vaccination intentions were weak, with 14.8% of respondents being unlikely to get vaccinated and another 23.0% unsure. Intent to vaccinate was highest for men, older people, and college-educated participants [21]. Furthermore, survey data collected from a total of 2343 Australian adults in April and August 2020 reported that willingness to vaccinate was higher in people with higher education, and willingness to vaccinate slightly decreased between April (87%) and August (85%) [11].

However, another research group conducted an anonymous cross-sectional survey among Chinese adults in March 2020. Among 2058 participants surveyed, 1879 (91.3%) stated that they would accept COVID-19 vaccination after the vaccine becomes available, among whom 980 (52.2%) wanted to get vaccinated as soon as possible, while others (47.8%) would delay the vaccination until the vaccine’s safety was confirmed [8]. In Japan a study was conducted in September 2020 that consisted of 1100 respondents. A significantly higher proportion of men were willing to get vaccinated (68.0%) than women (63.2%) The highest willingness to be vaccinated was found among the oldest age group (77.2%) and participants who had chronic diseases (78.4%) [22].

Unfortunately, so far no other similar studies have been conducted in the world that can be compared with our results. 

The limitations that characterize our study concern the poor representativeness of the sample for the oldest age group. Secondly, the study design was observational, which is useful mostly for generating hypotheses. Third, the vaccine reactions studied are mainly related to the Pfizer vaccine. Fourth, a small number of people took part in the study compared to the target group—people from group zero of the National Vaccination Program. On the other hand, the strengths of our work include the novelty of the topic, the large sample size and the demonstration of the scale of the problem as a result of the conducted research, which is the failure to report side effects after the vaccine is taken by people from the healthcare sector who are aware of the importance of reporting side effects. It can also be concluded that often people outside the health sector are not aware of the need to report side effects of vaccines. In addition, the study noted the frequent occurrence of certain side effects, such as pain at the injection site and fatigue. This publication should also send a signal to pharmaceutical companies to monitor vaccine side effects more closely. Another advantage is the share of 2% of the population of pharmacists in the study.

## 5. Conclusions

Persons who have previously had COVID-19 endured worse side effects after the first dose of the vaccine, while those who had not previously had COVID-19 felt stronger side effects after the second dose of the vaccine. Undesirable post-vaccination reactions may be the result of an individual reaction of the vaccinated person to the administration of the vaccine, a vaccine implementation error or vaccine administration error, or phenomena that are independent of vaccination and occur only by accident after vaccination. It is worth explaining in further research why such relationships occur.

In addition to this, our data confirm a low vaccination reported post-vaccination reactions to the Sanitary Inspection despite high frequency occurred. People in the health sector know about the ways and need to report vaccine reactions, while those outside the sector do not. It is important to remind people belonging to group zero of the National Vaccination Program about the need to report vaccine reactions, and to inform people from the next groups of the National Vaccination Program about possible ways of reporting side effects. Such activities will allow us to keep reliable statistics and Appendix A the Safety Data Sheets of vaccines against COVID-19.

## Figures and Tables

**Figure 1 vaccines-09-00502-f001:**
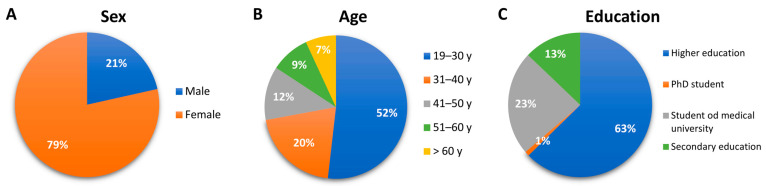
The characteristics of the study group: (**A**) sex, (**B**) age, (**C**) education.

**Table 1 vaccines-09-00502-t001:** Statistical dependencies from the survey.

Title	n	*p*
Side effects after receiving the vaccine—pain at the injection site after the first dose of the vaccine	1275	<0.0001
Side effects after receiving the vaccine—redness at the injection site after the first dose of the vaccine	696	<0.0001
Side effects after receiving the vaccine—pain in the limb after the first dose of the vaccine	766	<0.0001
Side effects after receiving second dose of the vaccine—a temperature above 38 °C	229	0.04
Association between the development of COVID-19 and the occurrence of stronger side effects after the first dose of the vaccine	Stronger first dose—103 Stronger second dose—40	*p* < 0.0001
Stronger side effects after the second dose of the vaccine in those who had not previously had COVID-19	Stronger first dose—298 Stronger second dose—768	*p* < 0.0001
Women suspected COVID-19 much more often than men	Women—238 Men—65	0.04
Women feel the need to vaccinate more often than men	Women—1243 Men—318	0.043
Women are more satisfied with the decision to vaccinate	Women—1251 Men—337	0.043
People with higher education knew the technology of the vaccine which they were vaccinated	1057	0.002
People with higher education choose a vaccine based on the VIRUS mRNA technology	Moderna 39 Pfizer 911	*p* < 0.0001
People with higher education showed need to be vaccinated against COVID-19	1057	0.036
Nurses expressed a strong desire to be vaccinated against COVID-19	33	NS

## Data Availability

The data presented in this study are available on request from the corresponding author.

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
