# Peer review of "The Incidence and Severity of Post-Vaccination Reactions after Vaccination against COVID-19"

_vaccines, 2021, doi:10.3390/vaccines9050502_

Round 1

Reviewer 1 Report

The topic presented in the study is of potential interest, but the methodology is not adequate. The aim of the study is not described in introduction. The methodology lacks of fundamental information (amongst which the description of the vaccine administrated, the number of persons invited to the survey, the questionnaire as appendix or additional material). Results are not clearly presented with respect to the aim of the study. In the discussion, limitations of the study are not sufficiently reported.

Author Response

I would like to thank the Reviewers for their time and comments.

I revised the manuscript according to the Reviewers comments.

  1. The topic presented in the study is of potential interest, but the methodology is not adequate. The aim of the study is not described in introduction.

This was corrected.

  1. The methodology lacks of fundamental information (amongst which the description of the vaccine administrated, the number of persons invited to the survey, the questionnaire as appendix or additional material).

This was corrected.

All this information is included in the Study Design, Population, and Sampling

  1. Results are not clearly presented with respect to the aim of the study.

This was corrected.

  1. In the discussion, limitations of the study are not sufficiently reported.

W dyskusji nie podano dostatecznie ograniczeń badania.

This was corrected.

Reviewer 2 Report

The paper reports the results of a survey of side effects experienced by individuals after the first and the second dose of vaccine against the SARS-CoV-2 (the ones developed by Oxford-AstraZeneca and Pfizer). A preliminary analysis of incidence and severity of symptoms related to the possibility of having been affected by the virus, diagnosed or not, communicated or not, is reported. The study has been performed by means of an anonymous questionnaire filled by the particular category of workers involved in the healthcare system. The results are presented giving the percentages of the different answers.

Test campaigns like the one here performed are always welcome for the researchers to have a larger set of data to analyze the phenomenon. In this sense, the paper can represent an interesting contribution as a data source.

A weakness of the paper is a poor discussion of the results and the absence of any comparative analysis with analogue studies, if any.

From a technical point of view, the sample is highly polarized, and the results cannot be easily extended to the generic population. It is clear that a better statistical sample is not available yet for countries that did not start the population vaccination early enough; however, the presented study can be considered as a preliminary result to be extended as soon as more data are available (for example, a larger and more uniform population involved in the questionnaire).

With the present data, the paper can be fruitfully improved adding some (at least one) comparative analysis with similar studies in different countries or for different population categories.

Some comments on the text structure are devoted to the presence of some typos, like the round bracket opened at line 73 of page 2 and never closed, and to some unclear paragraphs, like the ones in lines 169-180 and 181-187 of page 4, which seem both to refer to the side effects after the second dose of vaccine, like a reiteration, but with different numbers.

Author Response

I would like to thank the Reviewers for their time and comments.

I revised the manuscript according to the Reviewers comments.

  1. Test campaigns like the one here performed are always welcome for the researchers to have a larger set of data to analyze the phenomenon. In this sense, the paper can represent an interesting contribution as a data source. With the present data, the paper can be fruitfully improved adding some (at least one) comparative analysis with similar studies in different countries or for different population categories.

Unfortunately, so far no other similar studies have been conducted in the world that can be compared with our results. This information was added in the Discussion.

  1. Some comments on the text structure are devoted to the presence of some typos, like the round bracket opened at line 73 of page 2 and never closed, and to some unclear paragraphs, like the ones in lines 169-180 and 181-187 of page 4, which seem both to refer to the side effects after the second dose of vaccine, like a reiteration, but with different numbers.

This was corrected.

Reviewer 3 Report

The authors provided an analysis for 1964 response who received COVID 19 vaccination in Poland. The responses were collected form the health care workers in national immunization program stage 0. Overall, this paper is well written, the topic is interesting and timely. I suggested the accept for this manuscript. 

Author Response

I would like to thank the Reviewers for their time and comments.

I revised the manuscript according to the Reviewers comments.

I would like to thank the Reviewer for positive opinion

Round 2

Reviewer 1 Report

The Authors have addressed the previous comments.

Author Response

I would like to thank the Reviewer for their time.